# Let Offline RL Flow: Training Conservative Agents in the Latent Space of Normalizing Flows

## Abstract

Offline reinforcement learning aims to train a policy on a pre-recorded and fixed dataset without any additional environment interactions. There are two major challenges in this setting: (1) extrapolation error caused by approximating the value of state-action pairs not well-covered by the training data and (2) distributional shift between behavior and inference policies. One way to tackle these problems is to induce conservatism - i.e., keeping the learned policies closer to the behavioral ones. To achieve this, we build upon recent works on learning policies in latent action spaces and use a special form of Normalizing Flows for constructing a generative model, which we use as a conservative action encoder. This Normalizing Flows action encoder is pre-trained in a supervised manner on the offline dataset, and then an additional policy model - controller in the latent space - is trained via reinforcement learning. This approach avoids querying actions outside of the training dataset and therefore does not require additional regularization for out-of-dataset actions. We evaluate our method on various locomotion and navigation tasks, demonstrating that our approach outperforms recently proposed algorithms with generative action models on a large portion of datasets.

## 1 Introduction

Offline Reinforcement Learning (ORL) addresses the problem of training new decision-making policy from a static and pre-recorded dataset collected by some other policies without any additional data collection (Lange et al., 2012; Levine et al., 2020). One of the main challenges in this setting is the extrapolation error (Fujimoto et al., 2019) – i.e. inability to properly estimate values of state-action pairs not well-supported by the training data, which in turn leads to overestimation bias. This problem is typically resolved with various forms of conservatism, for example, Implicit Q-Learning (Kostrikov et al., 2021) completely avoids estimates of out-of-sample actions, Conservative Q-Learning (Kumar et al., 2020) penalizes q-values for out-of-distribution actions and others (Fujimoto & Gu, 2021; Kumar et al., 2019) put explicit constraints to stay closer to the behavioral policies.

An alternative approach to constraint-trained policies was introduced in PLAS (Zhou et al., 2020), where authors proposed to construct a latent space that maps to the actions well-supported by the training data. To achieve this, Zhou et al. (2020) use Variational Autoencoder (VAE) (Kingma et al., 2019) to learn a latent action space and then train a controller within it. However, as was demonstrated in Chen et al. (2022), their specific use of VAE leads to a necessity for clipping the latent space. Otherwise, the training process becomes unstable, and the optimized controller can exploit the newly constructed action space, arriving at the regions resulting in out-of-distribution actions in the original space. While the described clipping procedure was found to be effective, this solution is rather ad-hoc and discards some of the in-dataset actions which could potentially limit the performance of the trained policies.

In this work, inspired by the recent success of Normalizing Flows (NFs) (Singh et al., 2020) in the online reinforcement learning setup, we propose a new method called Conservative Normalizing Flows (CNF) for constructing a latent action space useful for offline RL. First, we describe why a naive approach for constructing latent action spaces with NFs is also prone to extrapolation error, and

then outline a straightforward architectural modification that allows avoiding this without a need for manual post-hoc clipping. Our method is schematically presented in Figure 1, where we highlight key differences between our method and the previous approach. We benchmark our method against other competitors based on generative models and show that it performs favorably on a large portion of the D4RL (Fu et al., 2020) locomotion and maze2d datasets.

## 2 PRELIMINARIES

**Offline RL** The goal of offline RL is to find a policy that maximizes the expected discounted return given a static and pre-recorded dataset $\mathcal{D}$ consisting of state-action-reward tuples. Normally, the underlying decision-making problem is formulated via Markov Decision Process (MDP) that is defined as a 4-elements tuple, consisting of state space $\mathbf{S}$, action space $\mathbf{A}$, state transition probability $p : \mathbf{S} \times \mathbf{S} \times \mathbf{A} \to [0, \infty]$, which represents probability density of the next state $s' \in \mathbf{S}$ given the current state $s \in \mathbf{S}$ and action $a \in \mathbf{A}$; bounded reward function $r : \mathbf{S} \times \mathbf{A} \times \mathbf{S} \to [r_{min}, r_{max}]$ and a scalar discount factor $\gamma \in (0, 1)$. We denote the reward $r(\mathbf{s}_t, \mathbf{a}_t, \mathbf{s}_{t+1})$ as $r_t$. The discounted return is defined as $R_t = \sum_{k=0}^{\infty} \gamma^k r_{t+k}$. Also, the notion of the advantage function $A(\mathbf{s}, \mathbf{a})$ is introduced – a difference between state-action value $Q(\mathbf{s}, \mathbf{a})$ and state value $V(\mathbf{s})$ functions:

$$
\begin{aligned}
Q^\pi(\mathbf{s}_t, \mathbf{a}_t) &= r_t + \mathbb{E}_\pi[\sum_{k=0}^{\infty} \gamma^k r_{t+k}] \\
V^\pi(\mathbf{s}) &= \mathbb{E}_{\mathbf{a} \sim \pi}[Q^\pi(\mathbf{s}, \mathbf{a})] \\
A^\pi(\mathbf{s}, \mathbf{a}) &= Q^\pi(\mathbf{s}, \mathbf{a}) - V^\pi(\mathbf{s})
\end{aligned}
\tag{1}
$$

**Advantage Weighted Actor Critic** One way to learn a policy in an offline RL setting is by following the gradient of the expected discounted return estimated via importance sampling (Levine et al., 2020), however, methods employing estimation of the Q-function were found to be more empirically successful (Kumar et al., 2020; Nair et al., 2020; Wang et al., 2020). Here, we describe Advantage Weighted Actor Critic (Nair et al., 2020) – where the policy is trained by optimization of log-probabilities of the actions from the data buffer re-weighted by the exponentiated advantage.

In practice, there are two trained models: policy $\pi_\theta$ with parameters $\theta$ and critic $Q_\psi$ with parameters $\psi$. The training process consists of two alternating phases: policy evaluation and policy improvement. During the policy evaluation phase, the critic $Q^\pi(s, a)$ estimates the action-value function for the current policy, and during the policy improvement phase, the actor $\pi$ is updated based on the current estimation of advantage. Combining all together, two following losses are minimized using the gradient descent:

$$
\begin{aligned}
L_\pi(\theta) &= \mathbb{E}_{(\mathbf{s}, \mathbf{a}) \sim \mathcal{D}}[- \log \pi_\theta(\mathbf{a}|\mathbf{s}) \cdot \exp(A_\psi(\mathbf{s}, \mathbf{a})/\lambda)] \\
L_{TD}(\psi) &= \mathbb{E}_{(\mathbf{s}, \mathbf{a}, r, \mathbf{s}') \sim \mathcal{D}}[(r + \gamma Q_\psi(\mathbf{s}', \mathbf{a}' \sim \pi_\theta(\cdot|\mathbf{s}')) - Q_\psi(\mathbf{s}, \mathbf{a}))^2]
\end{aligned}
\tag{2}
$$

Where $A_\phi(\mathbf{s}, \mathbf{a})$ is computed according to Equation 1 using critic $Q_\phi$ and $\lambda$ is a temperature hyperparameter.

**Normalizing Flows** Given a dataset $\mathcal{D} = \{x^{(i)}\}_{i=1}^N$, with points $x^{(i)}$ from unknown distribution with density $p_\mathbf{X}$ the goal of a Normalizing Flow model (Dinh et al., 2016; Kingma & Dhariwal, 2018) is to train an invertible mapping $\mathbf{z} = \mathbf{f}_\phi(\mathbf{x})$ with parameters $\phi$ to a simpler base distribution with density $p_\mathbf{Z}$, typically spherical Gaussian: $\mathbf{z} \sim N(0, I)$. This mapping is required to be invertible by design to sample new points from data distribution by applying the inverse mapping to samples from the base distribution: $\mathbf{x} = \mathbf{f}_\phi^{-1}(\mathbf{z})$. A full flow model is a composition of $K$ invertible functions $\mathbf{f}_i$ and the relationship between $\mathbf{z}$ and $\mathbf{x}$ can be written as:

$$
\mathbf{x} \xleftrightarrow{\mathbf{f}_1} \mathbf{h}_1 \xleftrightarrow{\mathbf{f}_2} \mathbf{h}_2 \cdots \xleftrightarrow{\mathbf{f}_K} \mathbf{z}
\tag{3}
$$

Log-likelihood of a data point $x$ is obtained by using the *change of variable* formula and can be written as:

$$
\begin{aligned}
\log p_\phi(\mathbf{x}) &= \log p_\mathbf{Z}(\mathbf{z}) + \log |\det(d\mathbf{z}/d\mathbf{x})| \\
&= \log p_\mathbf{Z}(\mathbf{z}) + \sum_{i=1}^{K} \log |\det(d\mathbf{h}_i/d\mathbf{h}_{i-1})|
\end{aligned}
\tag{4}
$$

Normalizing Flows models are optimized to directly maximize log-likelihood of data points using Equation 4.

# 3 METHOD

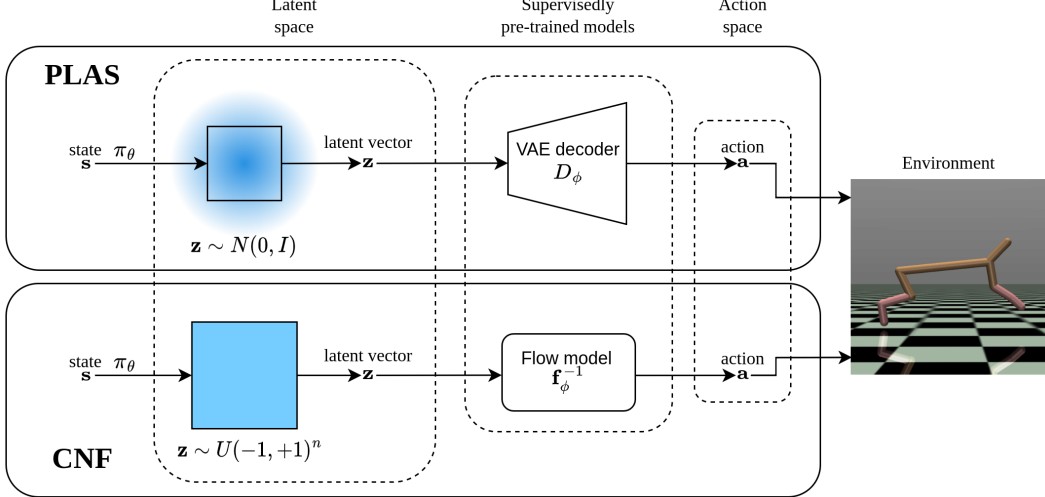

Figure 1: Schematic visualization and comparison of PLAS and CNF (ours) approaches. Both methods use an action encoder-decoder model trained in a supervised manner on an offline dataset and a controller model to select actions from the latent space of the encoder. PLAS algorithm uses VAE with normal latent distribution with unbounded support (represented as the blue circle) and restricts latent policy outputs to only a part of the latent space (represented as the black borders inside of latent space). Our algorithm uses Normalizing Flow instead of VAE and bound base distribution itself, allowing the latent policy to use the whole latent space.

In this section, we describe our approach named Conservative Normalizing Flows (CNF) in detail (Figure 1). We start by delineating the proposed architecture and then describe both the pre-training and policy optimization phases.

## 3.1 CONSERVATIVE NORMALIZING FLOWS

Normalizing Flows models are trainable bijective mappings, composed of fully invertible layers, that transform the original data space to the base distribution space. The latter is often referred to as the latent space. Typically, the base distribution is modeled as normal since it provides a tractable density and is easy to sample from (Kingma & Dhariwal, 2018). However, similar to the PLAS (Zhou et al., 2020), this distribution would result in unbounded support. Therefore, when learning a policy in the latent space to maximize q-values it may exploit the regions that lead to the out-of-distribution actions in the original space exaggerating the extrapolation error.

To illustrate this problem better, consider a toy 2-d task with modeling moons dataset which consists of points of two interleaving half circles (Buitinck et al., 2013) using normalizing flows. We train two normalizing flow models, first with the normal latent distribution and second with the uniform latent distribution. We additionally squeeze data points to lie in $(-1, +1)^2$ 2-d region via linear transformation in this experiment, and add the inverse of $\tanh$ function as the first layer of NFs models, so that $\tanh$ function applied as the last transformation during sampling from the model. Both trained NFs model the underlying training distribution well (Figure 2). We then select the NF model with normal latent space and gradually increase the amplitude $a$ of the latent samples drawn from $a \cdot N(0, I)$ during sampling new data points from the model, this process is presented in Figure 3. And, as expected, higher amplitude values result in more out-of-distribution data points.

One solution to avoid exploiting regions leading to out-of-distribution actions is to restrict policy output amplitude by some value. This approach was proposed in Zhou et al. (2020); Chen et al.

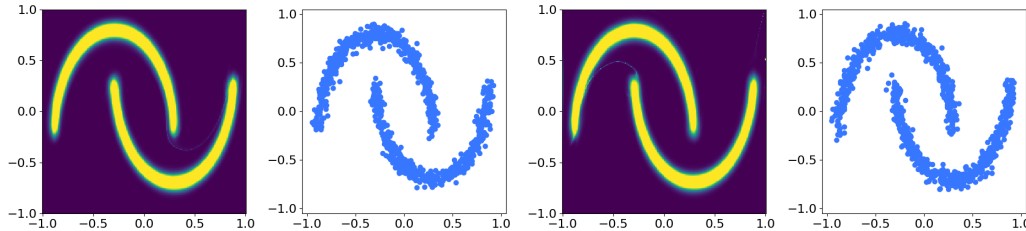

(a) NF-Normal, Density    (b) NF-Normal, Samples    (c) NF-Uniform, Density    (d) NF-Uniform, Samples

Figure 2: A toy example to demonstrate that both NFs either with Normal or Uniform latent distributions can recover the training data. However, as we demonstrated above, a potential controller trained in the latent space of normal-based NFs is still able to sample actions outside of the training dataset.

(2022), where the output of the deterministic policy in the latent space was modeled as $z = z_{max} \cdot \tanh(\pi_\theta(\mathbf{s}))$ and $z_{max}$ was set to 2.

However, it was shown that the optimal clipping value $z_{max}$ is different for every locomotion dataset in the D4RL benchmark (Zhou et al. (2020), Appendix C). Since $z_{max}$ value is essentially part of the latent space policy model, it is necessary to evaluate multiple values online after training to select the best one, which may not be feasible in some tasks.

To tackle this problem we would like to avoid this post-hoc clipping and construct a latent action space model that would prohibit the exploitation of out-of-distribution actions by design. To do so, we make use of NFs versatility, and add invertible $\tanh$ activation after the last layer of the Normalizing Flows model - it makes NFs outputs lie in a bounded $n-$dimensional[1] interval $(-1, +1)^n$, which in turn allows us to substitute normal base distribution with $n-$dimensional uniform $\mathbf{U}(-1, +1)^n$. With these changes, the potential actor model should not be capable of generating actions outside of the training distribution.

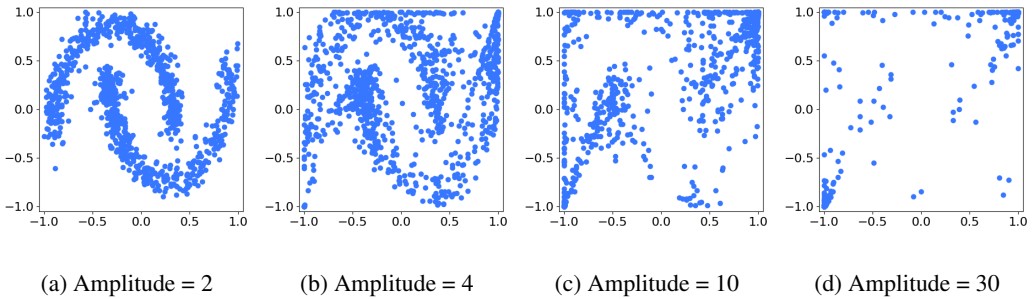

(a) Amplitude = 2    (b) Amplitude = 4    (c) Amplitude = 10    (d) Amplitude = 30

Figure 3: A toy example demonstrates that the normal-based NF can be manipulated by a controller to produce out-of-distribution training points. We model this by increasing the amplitude of the latent space samples, as we found this to happen in the preliminary experiments when the controller tries to maximize q-values. Note that the model with uniform latent space does not suffer from this problem because it already uses the whole latent distribution support during the training and sampling processes.

## 3.2 LATENT ACTION SPACE PRE-TRAINING

First, we start by pre-training conditional Normalizing Flow model $\mathbf{f}(\mathbf{a}|\mathbf{s})$ with parameters $\phi$ on the actions and states from the offline dataset, this is essentially a supervised learning problem. We use the same Normalizing Flow conditioning scheme as in the Singh et al. (2020) (for more details, see Figure 8 in the original paper): for each NF layer, we add additional input for the state vector. Given

---

[1] $n$ is the action space dimensionality.

a conditional NFs $\mathbf{f}$ with parameters $\phi$, we use it to compute log-likelihood $\log p_\phi(\mathbf{a}|\mathbf{s})$ of actions $\mathbf{a}$ conditioned on states $\mathbf{s}$ from the offline dataset $\mathcal{D}$, minimizing the following loss:

$$L_{\mathbf{f}}(\phi) = \mathbb{E}_{(\mathbf{s},\mathbf{a})\sim\mathcal{D}}[-\log p_\phi(\mathbf{a}|\mathbf{s})] \tag{5}$$

---

**Algorithm 1** CNF, Pre-training

---

1: **Input:** offline data replay buffer $\mathcal{D}$
2: Initialize flow action encoder parameters $\phi$
3: **repeat** until convergence
4:     Sample a mini-batch $B = \{(\mathbf{s}, \mathbf{a})\}$ from $\mathcal{D}$
5:     Compute flow loss using Equation 5, compute loss gradient and update flow model:

$$\nabla_{\theta_1} \frac{1}{|B|} \sum_{(\mathbf{s},\mathbf{a})\in B} [-\log p_\phi(\mathbf{a}|\mathbf{s})]$$

---

The final result of this optimization process is a mapping $\mathbf{f} : A \times S \to Z$, where $Z$ is the latent action space bounded by the $(-1, 1)$ interval. As this mapping is invertible, we can further transform latent vectors into the original action space for policy optimization.

## 3.3 POLICY OPTIMIZATION

The whole procedure is outlined in Algorithm 2. Here, we describe this phase in detail as follows. Given a pre-trained latent action space model, we freeze its parameters and add it as an action encoder during RL training. We modify policy optimization loss from Advantage Weighted Actor Critic (Nair et al., 2020) to use it with our flow model. Specifically, we train a stochastic latent policy model $\pi(z|s)$ with parameters $\theta$, which predicts $\mu$ and $\sigma^2$ for $\tanh(N(\mu, \sigma^2))$ distribution, and two critic models $Q_1(\mathbf{s}, \mathbf{a})$, $Q_2(\mathbf{s}, \mathbf{a})$ with parameters $\psi_1$ and $\psi_2$ to mitigate positive bias in the policy improvement step that is known to degrade the performance of value-based methods (Hasselt, 2010; Fujimoto et al., 2018). Note that our policy model operates in the latent space, not in the original space. By combining latent policy and a pre-trained normalizing flow model, we obtain the following loss function for policy optimization:

$$L_\pi(\theta) = \mathbb{E}_{(\mathbf{s},\mathbf{a})\sim\mathcal{D}}[\omega(\mathbf{s}, \mathbf{a}) \cdot |\mathbf{a} - \mathbf{f}^{-1}(z \sim \pi_\theta(\cdot|\mathbf{s})|\mathbf{s})|], \tag{6}$$

where weights $\omega$ comes from exponentiation of the Advantage function:

$$\begin{aligned}\omega(\mathbf{a}, \mathbf{s}) &= \exp\left(\left(Q(\mathbf{s}, \mathbf{a}) - Q(\mathbf{s}, \mathbf{f}^{-1}(z \sim \pi(\cdot|\mathbf{s})|\mathbf{s}))\right)/\lambda\right) \\ &= \exp\left(A(\mathbf{s}, \mathbf{a})/\lambda\right)\end{aligned} \tag{7}$$

and Q-function is set to the minimum between two trained models: $Q(\mathbf{s}, \mathbf{a}) = \min_{i=1,2} Q_i(\mathbf{s}, \mathbf{a})$, as proposed by Fujimoto et al. (2018). Here, $\lambda \in (0, \infty)$ is a temperature hyperparameter: for higher values training objective behaves similarly to behavioral cloning and for lower values it aims to maximize advantage. Overall, this loss function train a policy that maximizes the Q-values subject to a distribution constraint (Nair et al., 2020).

Together with the latent policy model, we optimize two critics with the standard Q-learning loss:

$$\begin{aligned}L_Q(\psi_1, \psi_2) &= \mathbb{E}_{(\mathbf{s},\mathbf{a},r,\mathbf{s}')\sim\mathcal{D}}[(Q_1(\mathbf{s}, \mathbf{a}) - y)^2 + (Q_2(\mathbf{s}, \mathbf{a}) - y)^2] \\ y &= r + \gamma\mathbb{E}_{\mathbf{a}'\sim\mathbf{f}^{-1}(\pi(\cdot|\mathbf{s}'))}[\min_{i=1,2} Q_i(\mathbf{s}', \mathbf{a}')]\end{aligned} \tag{8}$$

Since NFs are differentiable, we can compute the gradient of the given loss functions for the policy model weights using the chain rule and reparametrization trick (Kingma et al., 2019). Note, that one could bypass the differentiation through the NFs model, and optimize policy and critics in the latent space directly. However, we show in the Appendix (Section A.4), that this results in worse policies.

---

**Algorithm 2** Offline RL with CNF training

---

1: **Input:**  Pre-trained flow action encoder model $\mathbf{f}(\mathbf{a}|\mathbf{s})$ with parameters $\phi$, offline data replay buffer $\mathcal{D}$
2: Initialize actor $\pi$ with parameters $\theta$ and two critics $Q_1$ and $Q_2$ with parameters $\psi_1$ and $\psi_2$
3: **repeat** for a given number of train-ops
4:     Sample a mini-batch $B = \{(\mathbf{s}, \mathbf{a}, r, \mathbf{s}')\}$ from $\mathcal{D}$
5:     Sample next-state actions $\mathbf{a}'$ using policy $\pi(\cdot|\mathbf{s}')$ and flow $\mathbf{f}(\cdot|s')$ models, compute Q-target:

$$\mathbf{a}' = \mathbf{f}^{-1}(z \sim \pi(\cdot|\mathbf{s}')) \text{ for all } \mathbf{s}' \in B$$

$$y = r + \gamma(\min_{i=1,2} Q_i(\mathbf{s}', \mathbf{a}'))$$

6:     Compute critics loss using Equation 8, compute loss gradient and update models:

$$\nabla_{\psi_1,\psi_2} \frac{1}{|B|} \sum_{(\mathbf{s},\mathbf{a},r,\mathbf{a}') \in B} [(Q_1(\mathbf{s},\mathbf{a}) - y)^2 + (Q_2(\mathbf{s},\mathbf{a}) - y)^2]$$

7:     Sample actions $\mathbf{a}$ using policy $\pi(\cdot|\mathbf{s})$ and flow $\mathbf{f}(\cdot|s)$ models, compute advantage weights 7:

$$\hat{\mathbf{a}} = \mathbf{f}^{-1}(z \sim \pi(\cdot|\mathbf{s})|\mathbf{s}) \text{ for all } \mathbf{s} \in B$$

$$A(\mathbf{s}, \mathbf{a}) = Q(\mathbf{s}, \mathbf{a}) - Q(\mathbf{s}, \hat{\mathbf{a}})$$

$$\omega = \exp(A(\mathbf{s}, \mathbf{a})/\lambda)$$

8:     Compute policy loss using Equation 6, compute loss gradient and update policy model:

$$\nabla_{\theta_2} \frac{1}{|B|} \sum_{(\mathbf{s},\mathbf{a}) \in B} \exp(A(\mathbf{s},\mathbf{a})/\lambda) \cdot |\mathbf{a} - \hat{\mathbf{a}}|$$

---

## 4 EXPERIMENTS

To show how the proposed method works, we benchmark it on various locomotion and navigation tasks from the popular D4RL benchmark (Fu et al., 2020) comparing it to the other methods based on generative models - PLAS (Zhou et al., 2020) and LAPO (Chen et al., 2022). We also include AWAC (Nair et al., 2020) algorithm in our comparisons because we build our policy optimization method on top of it, and IQL (Kostrikov et al., 2021) algorithm because of its competitive performance across non-ensemble methods.

### 4.1 D4RL BENCHMARK

**Locomotion** We focus on three locomotion environments from the D4RL dataset: Walker2d-v2, Hopper-v2, and HalfCheetah-v2. For the Normalizing Flows model's pre-training phase, we divide the training dataset into two parts by separating $10\%$ portion of randomly selected data for validation. We run 50 experiments with the random search of hyperparameters from Table 3, and then select the best model according to the log-likelihood on the validation dataset for RL training. We train all models, including Normalizing Flows, latent policies, and critics, using Adam optimizer Kingma & Ba (2014). During the RL phase, we run 1 million training steps for actor and critic models on all environments except HalfCheetah-v2, where we use only $200.000$ training steps as it is enough for convergence. We evaluate the agent by running 10 episodes and averaging the scores over them once per 5000 (100 for HalfCheetah-v2) training steps. Hyperparameters for RL training are listed in Table 2.

For comparison, we implemented AWAC and IQL algorithms in our code base and used the publicly available PLAS implementation. We run AWAC and IQL algorithms with hyperparameters found in the papers Kostrikov et al. (2021); Nair et al. (2020); Zhou et al. (2020) and PLAS with hyperparameters recommended in Zhou et al. (2020). For LAPO, we use scores reported in the paper. The final scores of benchmarked algorithms are summarized in Table 1. Missing environments are labeled as '-' in the table.

To highlight the performance of our method, we include training curves in Figure 4. It can be seen that the proposed algorithm exhibits preferable performance on all 9 locomotion datasets, especially on the HalfCheetah-v2 environment.

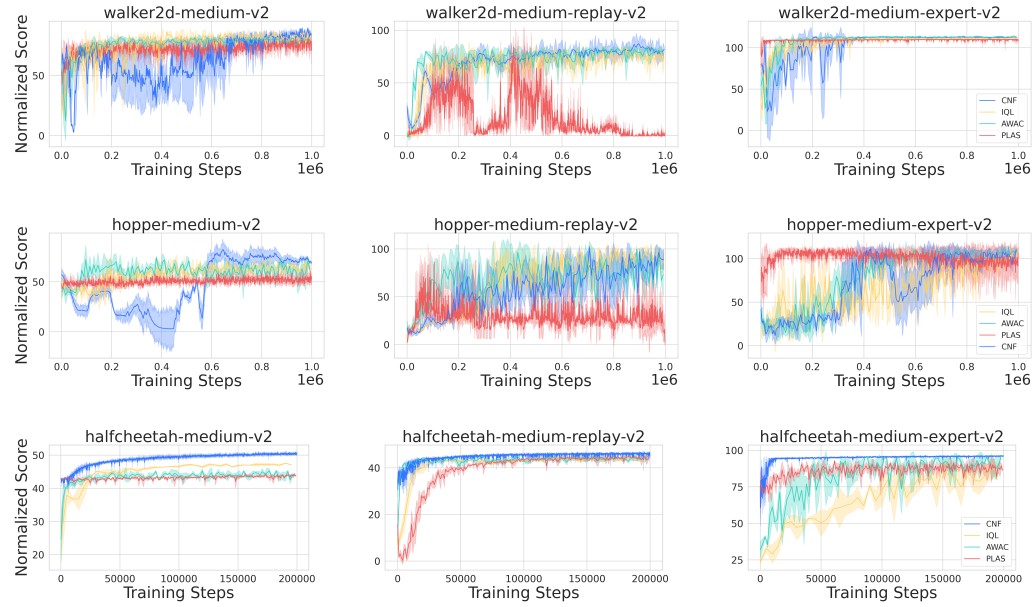

Figure 4: Average normalized performance on D4RL locomotion tasks. The x-axis denotes the training steps. Each curve is averaged over 3 random seeds. Shaded area represents one standard deviation.

**Maze2d** For our next experiment, we choose three maze2d datasets with increasing layout complexity: umaze-v1, medium-v1, and large-v1. We use the same flow pre-training scheme as before: we run 20 experiments with random hyperparameters from Table 3 and select the model with the highest log-likelihood on the validation dataset. We list the hyperparameters for RL training used in this experiment in Table 2. For comparison, we use our implementation of IQL, rely on a publicly available PLAS implementation, and report scores from the LAPO paper. As can be seen in Table 1, our method outperforms baselines in 2 out of 3 environments.

## 4.2 ABLATIONS

To study the importance of each major component in the proposed CNF method, we conduct additional ablation experiments. We begin by comparing training policies in uniform and normal latent spaces. Then, we examine additional clipping for the latent policy to see if it can operate in normal latent space. And finally, we integrate the VAE model into our approach and compare performance between Normalizing Flows and VAEs for training policies in latent spaces.

**Normalizing Flows with normal latent distribution, no clipping** In the first experiment, we compare the performance of the proposed method but with normal latent distribution in Normalizing Flow and latent policy models. We use a conventional Normalizing Flows encoder without $\tanh$ activation after the last layer and with the normal latent distribution. We pre-train action encoders in the same way as before, using only the best hyperparameters from previous experiments (note that these hyperparameters result in very similar models in terms of the training and validation losses, as depicted in Figure 8). We also remove $\tanh$ activation after the last layer from the latent policy model, letting it operate over the whole latent space of the pre-trained Normalizing Flow action encoder, which we make to predict $\mu$ and $\sigma^2$ for $N(\mu, \sigma^2)$ distribution. We again select the HalfCheetah-v2 environment and run 3 experiments with different random seeds per dataset. We plot the results in Figure 5. One can see that the performance degraded substantially, indicating that latent policy without clipping could not be trained to produce a competitive performance.

Table 1: **Normalized performance on D4RL benchmark.** The scores are averaged over 10 final evaluations and 3 random seeds. The results are reproduced for all of the algorithms except LAPO, for which we take the values stated in the original paper. CNF outperforms other methods on 8 out of 9 locomotion datasets, and on two out of three maze2d datasets.

| Environment | AWAC | IQL | PLAS | LAPO | CNF |
|---|---|---|---|---|---|
| walker2d-medium-v2 | 78.20 | 78.80 | 71.2 | 80.75 | **83.60** $\pm$ 3.01 |
| walker2d-medium-replay-v2 | 76.76 | 74.49 | 2.74 | - | **81.96** $\pm$ 1.98 |
| walker2d-medium-expert-v2 | **112.97** | 111.96 | 108.13 | - | 112.32 $\pm$ 0.21 |
| hopper-medium-v2 | 62.59 | 63.7 | 52.93 | 51.63 | **69.32** $\pm$ 1.04 |
| hopper-medium-replay-v2 | 73.12 | 87.72 | 3.17 | - | **89.04** $\pm$ 10.39 |
| hopper-medium-expert-v2 | **109.64** | 109.05 | 106.5 | - | 108.6 $\pm$ 5.45 |
| halfcheetah-medium-v2 | 43.15 | 47.4 | 43.78 | 45.97 | **50.55** $\pm$ 0.53 |
| halfcheetah-medium-replay-v2 | 42.00 | 43.2 | 44.8 | - | **45.84** $\pm$ 0.31 |
| halfcheetah-medium-expert-v2 | 87.40 | 78.95 | 86.63 | - | **96.23** $\pm$ 0.20 |
| locomotion-v2 average | 76.20 | 77.25 | 57.76 | - | **81.94** |
| maze2d-umaze-v1 | - | 37.69 | 53.9 | 118.9 | 62.9 $\pm$ 10.36 |
| maze2d-medium-v1 | - | 35.45 | 66.4 | 142.8 | **155.89** $\pm$ 10.49 |
| maze2d-large-v1 | - | 49.64 | 107.2 | 200.6 | **212.8** $\pm$ 2.23 |

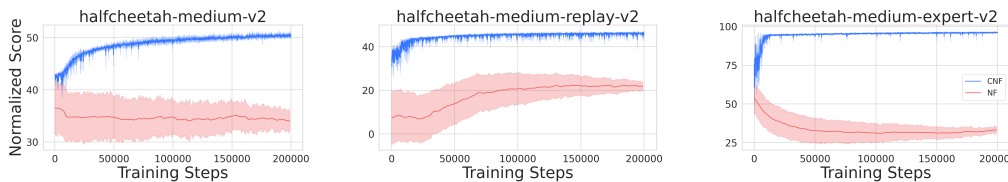

Figure 5: Comparison of the proposed method with uniform (CNF) and normal (NF) latent spaces. Policy performance is significantly worse when the latent space is normal.

**Normalizing Flows with normal latent distribution, manual clipping** As it was shown in the previous ablation, latent policies without clipping in normal space perform poorly. To test how different clipping values affect the agent's performance, we did the following experiment, which is similar to the previous one, except we manually clip latent policy output by some value. To do so, we return tanh activation at the end of the policy model and multiply it by $a$, treated as a hyperparameter. Latent policy output is modeled as $z \sim a \cdot \tanh(N(\pi_\theta(.|s)))$, where $N$ is a normal distribution with parameters predicted by the policy model. This experiment is similar to Zhou et al. (2020) Ablation C, but instead of VAEs, we use Normalizing Flows to extract latent policies using Algorithm 2. Also, we use datasets from the Walker2d-v2 environment. We examine several values for parameter $a$ and compare them with the proposed method. The results are averaged over three random seeds and are shown in Figure 6. One can see that optimal clipping values are different for each dataset: for the medium dataset, there is no disparity in performance, but the clipping value of 3 produces slightly better results and almost matches the performance of the uniform latent space; for the medium-replay, it equals to 2 and for the medium-expert, it equals to 1. On the other hand, CNF, parameterized with the uniform latent distribution, does not add extra clipping value as a hyperparameter and performs better on each dataset.

**Action encoder model: Normalizing Flow and VAE** In this experiment, we compare the performance of latent policies obtained by the CNF with different action encoders, namely, Normalizing Flow and VAE. We adopt the best VAE architecture and training parameters from Zhou et al. (2020), and then integrate pre-trained models into the latent policy training as in Algorithm 2. For this comparison, we rely on the HalfCheetah-v2 environment and plot training curves for PLAS, CNF, and CNF-VAE. Results are presented in Figure 7. One can see that the CNF performs best with the Normalizing Flow action encoder. On the other hand, the use of the VAE encoder shows performance

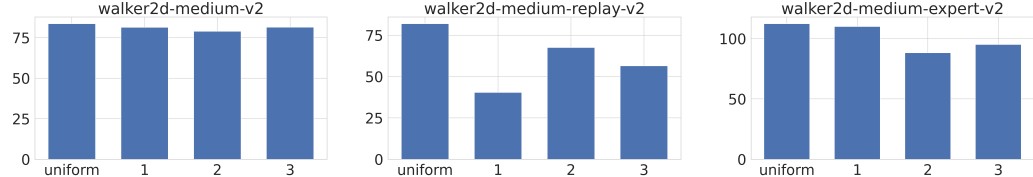

Figure 6: Normalized average performance of the proposed method (uniform on the X-axis) and latent policies with clipped normal latent distribution. Number on the X-axis is the clipping value. Optimal clipping value is different for each dataset and final performance is better for the proposed method.

similar to PLAS. On the halfcheetah-medium-replay-v2 dataset, it starts from a commensurate performance to CNF, but exhibit marginal improvement during the training process. This experiment indicates the importance of the whole pipeline with the use of Normalizing Flows as opposed to the VAEs.

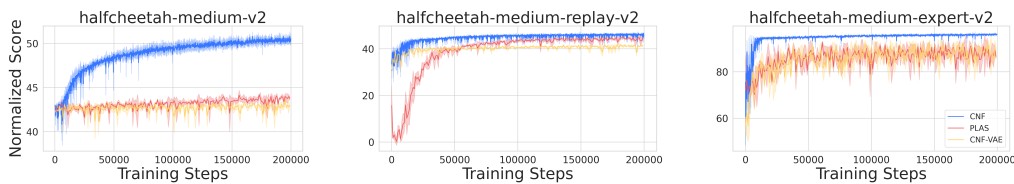

Figure 7: Comparison of CNF, CNF with VAE (CNF-VAE) and PLAS. CNF with Normalizing Flow and uniform distribution performs above all. For normal distribution optimal clipping value is different for each dataset.

## 5 CONCLUSION

In this work, we presented a new deep offline RL method called Conservative Normalizing Flow (CNF). It constructs a latent action space with the use of the NFs model and then runs a policy optimization within it. This approach makes trained policies capable of fully utilizing the latent space as opposed to the post-hoc manual clipping procedures in PLAS Zhou et al. (2020) and LAPO Chen et al. (2022). We benchmarked our method against other competitors based on generative models and showed that it performs favorably on a large portion of D4RL Fu et al. (2020) locomotion and maze2d datasets.

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

# A   APPENDIX

## A.1   RELATED WORK

**Normalizing Flows in RL** Several prior works applied Normalizing Flows models for reinforcement learning tasks. In the work Haarnoja et al. (2018) Normalizing Flows were used for training hierarchical policies using the Soft Actor-Critic RL algorithm. They choose Normalizing Flows because it provides an expression for exact likelihood computation and has an intuitive way to stack a sequence of models to construct one hierarchical model. In contrast to our work, they did not modify NF's latent space, they did not use offline pre-training of flow models and train models in the online RL framework.

The PARROT work (Singh et al., 2020) proposes to use the Normalizing Flows action encoder during the behavioral cloning pre-training phase before running reinforcement learning on the target task. After pre-training, they freeze Normalizing Flow and run reinforcement learning on a new and unseen task. They aimed to develop a method leveraging near-optimal demonstrations during Normalizing Flows pre-training and speed up the convergence of RL by better and meaningful exploration, while we use Normalizing Flows in purely offline RL setting and to make agent conservative.

**Offline RL** A large portion of recently proposed deep offline RL algorithms focuses on addressing the extrapolation issue, trying to impose a certain degree of conservatism limiting the deviation of the final policy from the behavioral one. Researchers approached this problem from multiple angles. For example, Kumar et al. (2020) proposed to directly penalize out-of-distribution actions, while Kostrikov et al. (2021) avoids estimating values for out-of-sample actions completely. Others Fujimoto & Gu (2021); Kumar et al. (2019); Jaques et al. (2020) put explicit constraints to stay closer to the behavioral policy. Here, we took a different approach by constructing a latent action space that allows us to bypass the need for explicit regularizations.

**Offline RL with generative models** In the method named PLAS (Zhou et al., 2020), the authors proposed to pre-train conditional variational autoencoder on actions from an offline dataset. This idea resembles ours, but to make agent conservative authors restrict policy outputs in the latent space to a fixed range. In our work, we aimed to make a better action encoder model by switching from VAEs to Normalizing Flows. This allows us to utilize the whole latent space of the action encoder, avoid manual clipping, and we experimentally demonstrate that our approach leads to better performance on the popular offline RL benchmark.

One recent approach called LAPO (Chen et al., 2022) proposes to train an action encoder together with a reinforcement learning agent. They motivated this by observing that the action distribution does not match the return distribution in the training data set, and therefore actions that lead to higher returns are more important for action encoder training. In our work, we examine an orthogonal direction, studying a different generative model for action encoding.

## A.2 HYPERPARAMETERS

Table 2: Hyperparameters for latent policy and critics models for RL training phase on locomotion and maze2 tasks. We ran a grid-search for the values written in square brackets. Rest of the parameter were fixed for all of the datasets.

| Hyperparameter | value |
| --- | --- |
| number of training steps | 1000000 |
| number of training steps (HalfCheetah-v2) | 200000 |
| number of layers (locomotion) | [3, 4] |
| number of layers (maze2d) | 4 |
| hidden size | 256 |
| learning rate | 3e-4 |
| batch size (locomotion) | [256, 512, 1024] |
| batch size (maze2d) | 10240 |
| $\lambda$-temperature (locomotion) | 1/3 |
| $\lambda$-temperature (maze2d) | 1/10 |

Table 3: NFs hyperparameters for the supervised pre-training phase on locomotion and maze2 tasks. Medium, medium-replay, and medium-expert datasets are marked as m, m-r, and m-e correspondingly. We sample learning rate and weight decay from the continuous uniform distribution.

| Hyperparameter | value |
| --- | --- |
| number of training steps | 100000 |
| number of layers (locomotion m and m-e datasets) | 12 |
| number of layers (locomotion m-r and maze2d datasets) | 4 |
| hidden size (locomotion m and m-e datasets) | 256 |
| hidden size (locomotion m-r and maze2d datasets) | 64 |
| learning rate | min = 1e-5, max = 3e-3 |
| weight decay | min = 0.0, max = 1e-2 |
| batch size | [512, 1024, 2048] |

## A.3 NORMALIZING FLOWS TRAINING CURVES

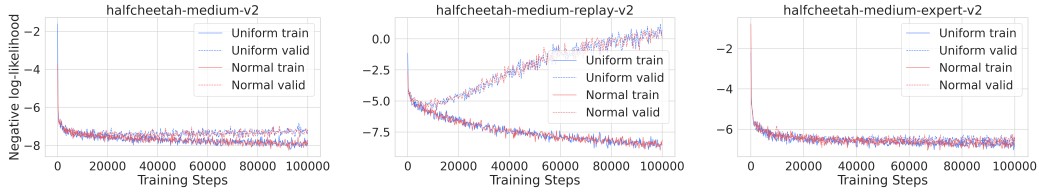

Figure 8: Training and validation loss for Normalizing Flows pre-training with Uniform and Normal latent spaces. Both training and validation curves are almost identical between models with different latent spaces, which means that models have similar encoding and reconstruction quality. For each latent policy training, we select NF model with the lowest validation metric.

### A.4 TRAINING CONTROLLER IN LATENT SPACE

Ideologically, policy optimization can be carried out directly in the latent space. This can be done by simply substituting the original actions with their latent counterparts. After this substitution, a myriad of offline RL algorithms can be used to extract a new policy. To test if this is a viable approach, we train actor and critic models in the latent space of the Normalizing Flows without utilizing gradients from the action encoder during the policy optimization phase. First, we convert actions from the original environment's action space to the latent space by encoding them with the pre-trained action encoder. After that, we run the AWAC algorithm with no additional changes to the dataset with converted actions. We conduct this experiment on HalfCheetah-v2 datasets: medium, medium-replay, and medium-expert. Results are presented in Figure 9. It can be seen that this approach shows promising results, but convergence speed and the final score are slightly lower on all datasets. We conjecture that the performance is lower because a metric in the latent space is induced from the original action space by the flow model, and optimization of the distance between latent vectors corresponds to optimization of the distance between actions only implicitly. When training policy using metrics from the original action space, as suggested by Equation 6, the Normalizing Flows model provides gradients to the policy model that guide it to minimize the actual distance between actions.

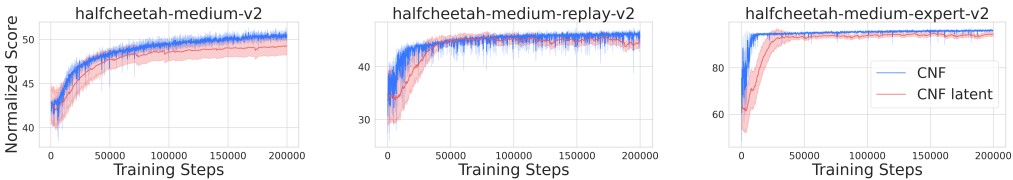

Figure 9: Comparison of CNF trained in original (blue) and latent (red) action spaces. Final performance is slightly lower for the training directly in latent space.

