# OpenReview forum: "Let Offline RL Flow: Training Conservative Agents in the Latent Space of Normalizing Flow"
_ICLR.cc/2023/Conference — Submitted to ICLR 2023_

### Official Review · Reviewer_DZCL · 2022-10-17

**Confidence:** 4
**Correctness:** 3
**Technical Novelty And Significance:** 3
**Empirical Novelty And Significance:** 2
**Recommendation:** 5

**Clarity, Quality, Novelty And Reproducibility:**

Clarity: very poor.

Quality: poor.

Novelty: very good.

Reproducibility: Questionable, even if some of the hyper parameters are provided, but I do not see any statement about code release.

**Strength And Weaknesses:**

Strength:

- The core idea of using NF is quite interesting and might be a good replacement for PLAS.

Weakness:

- The paper is too immature with numerous mistakes and quite poor presentation. In the current form it is far from acceptable.

**Summary Of The Paper:**

The paper utilized *Normalizing Flows* models, which are trainable bijective mappings, composed of fully invertible layers. The core idea is to obtain a bounded support in the latent space by design via shifting the base distribution space (that of the latent variables) from a normal distribution to a bounded uniform one. The proposed method seems to be based on PLAS (Zhuo et al.), which constructs a latent space using VAE that maps to the actions which are well-supported by the training data. It then learns optimal policy in this space.

**Summary Of The Review:**

I had a hard time following the paper. It contains too many mistakes, inconsistency, and unclear/confusing parts. I would strongly recommend the authors should start fresh and completely rewrite the draft.

Here are my main comments; hopefully they help:

- The choice of $f(x) = tanh(x)$ as the activation needs to be discussed. In particular, $tanh$ has a strong saturation for when $x$ passes one. In other words, its behaviour is *almost* linear for when the advantage function is in $[-1,1]$ and somehow clipping $A(s,a)$ when falling outside unit interval (or $[-2,2]$).  Is it a limitation? Discussion needed.

- In equation 2, $\textbf{f}$ applies on the advantage function, while later $\textbf{f}$ is defined over $(s,a)$. If by these you mean the same function, fix your definitions. If not, then use different letters.

- Section 3 is quite unclear. Specially the paragraph before Fig. 3. Further, the range of data (from -1 to 1 on each axis) seems to be arbitrary with no explanation provided.

- Page 4, end of first paragraph: “one is still able to find latent vectors that would result in out-of-distribution points.”  -> it is not inferred from the figures. Perhaps consider an example in the figures for this statement.

- The idea of mapping the value function and training the Q network in the mapped space has also been studied in the recent literature. Though it might not be directly related, it could be beneficial to borrow some ideas, or proofs. See for example: https://arxiv.org/abs/2203.07171

- Equation 6 is very confusing: sampling from $\pi(\cdot |s)$ gives rise to actual actions (not the z space).

- Consider discussing equations 6-9 in much more details, as they are the core of your paper.


Other comments:

- Figure 1 seems to never have been referred to in the paper. Also, Figure 2 is referred after Figure 3.

- When citing a paper indirectly (many times throughout the manuscript), use “\citep” rather than “\cite” to include parentheses outside both the names and the date.

- Your definition of reward is indeed the expected reward (over next state).

- You allow for $\gamma = 1$ and at the same time you define return on an infinite horizon, which may become unbounded. Either remove the “no discount” scenario, or rather make your MDP episodic by introducing terminal states and necessitating termination.

- Equation 1 should be the definition of advantage function (it is misplaced).

- Your notation is not consistent. From what I see, you use CAP-BOLDFACE for sets, but then later you also use it for random variables X and Z. Similarly for the lower-case letters. Please consider a full overhaul of your notations.

- Section "Normalizing Flow": X and Z are random variables, NOT distributions.

- In section “Critic Regularized Regression” you used script-D for the dataset, but in the next part you used D. Again, I strongly recommend consistency in your notation to help your readers following the concepts better.

- Figure 1 -> in the CNF graph, $z\sim U(-1,1)^n$. Therefore, the color should be a solid blue instead of radial gradient.

---

> ### Author Response · Authors · 2022-11-19
> **Response**
>
> Thanks for the feedback and manuscript suggestions, they are extremely helpful to identify the confusing parts. We agree that the original version of the paper contained some important typos and misleading notations. We addressed all of your points in the updated version of the paper, hopefully, now it should make less confusion. Moreover, we added more explanations for particular algorithmic choices and few important ablations (e.g., NF-Normal with clipping versus CNF).
>
> As for the reproduction, we also uploaded the source code as a supplementary material.

---

### Official Review · Reviewer_gFEy · 2022-10-24

**Confidence:** 3
**Correctness:** 2
**Technical Novelty And Significance:** 3
**Empirical Novelty And Significance:** Not applicable
**Recommendation:** 5

**Clarity, Quality, Novelty And Reproducibility:**

If the major performance improvement is latent space clipping ([-1,1]), the novelty of the proposed method is very limited.



**Strength And Weaknesses:**

This paper is generally well-written and easy to follow.
It is reasonable that the uniform distribution is better than the normal distribution for the latent space in ORL.

The reviewer has two concerns as follows.
First, the necessity of the flow model is not well supported.
Figure 6 presents that the proposed model performs poor scores when using the normal distribution as the latent space.
Please report an ablation study for the pre-trained models (VAE decoder vs. flow model).

Second, the performance improvement by the latent space clipping is missing.
The latent space with the uniform distribution has clipped values in [-1,1], while the latent space with the normal distribution is unbounded.
Normal distribution with value clipping in [-1,1] will provide fair comparisons.
In this regard, the reviewer wants to see the results of NF-uniform under the experiment setting in Figure 3.
If the results are the same as in Figure 2(b) regardless of the value of amplitude, the proposed method does not address "extrapolation".




**Summary Of The Paper:**

This paper presents a method using normalizing flows for offline reinforcement learning (ORL).
To alleviate the difference between training and testing datasets, the proposed method maps uniform distributions to action spaces using the flow model.
Experimental results present that the proposed method outperforms the state-of-the-art methods in many benchmark datasets.


**Summary Of The Review:**

The proposed method is interesting and the paper is well-written.
However, some ablation studies and clarifications of the performance improvements are missing.

---

> ### Author Response · Authors · 2022-11-19
> **Response**
>
> Thank your for the feedback. We address the mentioned concerns as follows.
>
> > First, the necessity of the flow model is not well supported. Figure 6 presents that the proposed model performs poor scores when using the normal distribution as the latent space. Please report an ablation study for the pre-trained models (VAE decoder vs. flow model).
> >
>
> We added this ablation into the main text, observing that the CNF outperforms CNF-VAE (which is essentially AWAC + VAE) by a considerable margin (see Figure 7)
>
> > Second, the performance improvement by the latent space clipping is missing. The latent space with the uniform distribution has clipped values in [-1,1], while the latent space with the normal distribution is unbounded. Normal distribution with value clipping in [-1,1] will provide fair comparisons.
> >
>
> Thank you, this is indeed an important experiment that we missed in the original version of the paper. In the updated version, we added the experiments with clipped NF-Normal (see Section 4.2, and Figure 6). Notably, CNF outperforms NF-Normal for a range of clippings (including [-1, 1]).

---

### Official Review · Reviewer_17pi · 2022-10-24

**Confidence:** 4
**Correctness:** 4
**Technical Novelty And Significance:** 2
**Empirical Novelty And Significance:** 3
**Recommendation:** 6

**Clarity, Quality, Novelty And Reproducibility:**

-

**Strength And Weaknesses:**

Overall the paper is written well and is intuitive. The use of normalizing flows to construct a latent space that leads to conservative agents by design is unique yet neat. The ability to learn better action-spaces that allows to model the state-action coverage of the dataset by default can be very useful for a many downstream and RL learning problems.

It would be wonderful to see some experimental analysis on the qualitative or quantitative differences between the action sampling portion of the current model and other methods in terms of being able to sample state action pairs that are well covered in the data set along with if and when these models are prone to over-fitting as discussed in the paper. Some empirical evaluation of different action generating functions in real wold datasets would be really helpful to push the community towards using Normalizing Flows with uniform latent distribution as a principled way of sampling actions in offline RL settings.

Regarding the experimental results, while it is competitive with the SOTA methods in many cases, it is not trivial why it fails to perform well when it does not. For example it is surprising that the method does not work as well for domains like maze2d umaze and medium expert tasks. (It is expecially surprising that it is performing worse in hopper medium expert as compared to hopper medium replay. This is an anomaly as generally  most offline RL algorithms tend to perform better with expert dataset. I would love to know more about what authors think about this result. Moreover it would be wonderful to have results on more involved domains such as antmaze to further bolsterd the claims made in the paper.

**Summary Of The Paper:**

This paper introduces Conservative Normalizing Flow (CNF) which encourages the final offline RL policy to remain withing the data distribution while being able to run policy optimization over an unconstrained latent space. They achieve this by pre-training the latent space to be a uniform distribution using normalizing flows that learns bijective mappings using fully invertible layers. The key novelty here is the use of bounded functions such as tanh at the last layer of normalizing flow such that the latent distribution may be learned as a uniform distribution between -1 and +1. This allows us to model/sample the behavioral action distribution using the entire latent space. (no clipping required.). Experimental results show that they are competitive with other SOTA offline RL methods in many domains.

**Summary Of The Review:**

Overall the paper provides a novel approach on sampling well covered actions for the downstream task of offline RL, However the experimental results show some anomalies that I would like to be been elaborated further by the authors. I would be willing to change my score with some explanations regarding the anomalies in the scores and some additional experiments.

---

> ### Author Response · Authors · 2022-11-19
> **Response**
>
> Thank you for the feedback. We updated the scores for the locomotion tasks, now the proposed method outperforms other latent-policy approaches on the whole set. The issue was the use of three layers for the policy network as opposed to typically utilized four layers in other algorithms (this is actually common for offline RL algorithms, e.g., CQL with 3 layers instead of 4 shows way worse results). As for the additional experiments, unfortunately, we do not possess an access to the real-world problems (e.g. robot hardware, as in the original PLAS paper). However, to further bolster the claims on manual clipping and the efficiency of latent-space utilization, we added more ablations with manual clipping of NF-Normal, demonstrating the preference of the uniform latent space.

---

### Official Review · Reviewer_iyr1 · 2022-10-24

**Confidence:** 5
**Correctness:** 3
**Technical Novelty And Significance:** 2
**Empirical Novelty And Significance:** 2
**Recommendation:** 6

**Clarity, Quality, Novelty And Reproducibility:**

**Clarity:**
For the most part, the paper is written clearly. In particular, the abstract and intro motivate the problem well, and clearly outline the paper's contributions.
- Figure 1 nicely explains both the approach and its difference from prior work.
- The toy example in Figure 3 helps build intuition for the problem.

However, the paper becomes less clear in the methods section. More details for Section 3.3 would be helpful. For example, providing an explanation of algorithm 2 in the text would be more clear. There is far too much detail of low-level hyperparameter settings in Section 4.1 that could be moved to the Appendix, in order to spend more time in Section 3 explaining the methodological choices and the actual algorithm. Other issues with the methods clarity:
- The ordering of presentation of the equations seems unclear. A critic is needed to estimate Eq 7 but hasn't been introduced yet
- Why use two critic networks as in Fujimoto? This should be motivated better.
- There appears to be a typo in step 7 of algorithm 2, where it takes A(A(s,a)). What is the advantage of the advantage?

Minor clarity issues:
- "Supervisely" pre-trained models in Figure 1
- Why use the normalizing flow conditioning scheme from PARROT? Why do you hypothesize it will be useful here / how is it relevant?

**Quality:**
- This paper criticizes prior work for having to manually bound the RL agent's action space with the VAE approach, but also takes the approach of manually bounding the agent's actions to be within the (-1,+1) interval of support of the uniform distribution. So it would be better to tone down or rephrase those claims. For example, the related work talks about how CNF "makes agents conservative by design, rather than squeezing policy outputs". But in effect you are squeezing policy outputs also.
- It appears the paper tests on a subset of the D4RL environments. Why were these chosen vs. others? This should be justified better.
- As mentioned above, according to the results CNF helps in 67% of environments, but it hurts badly in others. Are these results of significant interest to the community?
- The ablation studies such as those of Figure 6 are a great thing to include and help illustrate why the approach is useful. However, the top paragraph of p. 8 states that the ablation using a Normal latent space was carried out with the best hyperparameters found for the Uniform latent space. So how do we know that the difference in performance can't be attributed to doing better hyperparameter tuning for the proposed approach?

**Originality:**
The idea of pre-training a conditional latent model on data and then using an RL agent to pick embedding vectors was proposed by Zhou et al. (2020). The difference with this work is that while Zhou used a VAE, this work proposed to use normalizing flows. This is a relatively minor change from Zhou et al.

The paper is missing a relevant citation to https://arxiv.org/abs/2010.05848, which is a conservative Offline RL method that minimizes divergence from the behavior policy using KL-control.

**Strength And Weaknesses:**

A strength of the paper is that it precisely targets a clear hypothesis: whether NF can improve the ability for an offline RL agent to learn to control a pretrained model vs. a VAE. The motivation for the paper is clear, as is the distinction from prior work.

A weakness of the paper is its significance, or impact. It makes a relatively minor tweak to the approach proposed by Zhou et al. 2020, and while the results demonstrate gains on 67% of environments sampled for the paper (8/12), it also leads to dramatically worse performance in some environments. Thus, the novelty and effectiveness of the approach is limited.

One possible avenue for future work that the authors could consider in order to increase the significance of their results, is to think about applying this approach to learning to control large language models with RL. Currently, it's difficult to train LLMs with RL due to the complexity of LLM infrastructure and training with RL on such a large scale (i.e. fine-tuning all the parameters could be prohibitively expensive). One approach would be to impose a bottleneck embedding layer within an LLM (see e.g. https://arxiv.org/abs/2209.06792), and then train an RL agent to pick embedding vectors to optimize some language objective. The issue is that this can be quite difficult to get working. The linked vec2text paper suggests that using a VAE doesn't work well in this context. So perhaps normalizing flows could be useful there as well.

**Summary Of The Paper:**

Offline RL is challenging when the learned RL policy drifts too far from the support of the dataset. Thus, many offline RL methods use some form of constrained or conservative policy update to ensure that the RL policy remains close to the behavior policy of the dataset. One method for doing this is to train a generative model (e.g. a VAE) on the data, and learn an RL policy that picks embedding vectors which are decoded into actions by the VAE decoder. This paper proposes a slight tweak on that prior approach: rather than learning a VAE on the dataset, what if we learned a model using normalizing flows? Using this Conservative Normalizing Flows approach, the authors transform the prior into a Uniform distribution rather than a Normal distribution, which ensures that the RL policy can choose samples anywhere within the support of the Uniform distribution without generated OOD samples, unlike with the VAE. The results show some improvement over existing methods on a sample of tasks from the D4RL offline RL benchmark.

**Summary Of The Review:**

In summary, the paper clearly and precisely investigates a specific hypothesis related to using normalizing flows rather than a VAE for a particular type of Offline RL approach. It is benchmarked against relevant Offline RL baselines and shows an improvement in 67% of environments, and a strong detriment in others. Relevant ablations are conducted to give intuition as to why the proposed approach is important. The novelty above prior work is somewhat limited.

---

> ### Author Response · Authors · 2022-11-19
> **Response**
>
> Thank you for the feedback. We address your concerns as follows.
>
> ### On typos, clarity, and explanations.
>
> We significantly revised the manuscript by carefully removing ambiguous notations and fixing typos. Moreover, we added more reasoning behind algorithmic choices, hopefully, these now should make more sense. Also, thanks for pointing us to the missing citation, we added it into the related work.
>
> ___
> ### On other comments
>
> > A weakness of the paper is its significance, or impact. It makes a relatively minor tweak to the approach proposed by Zhou et al. 2020, and while the results demonstrate gains on 67% of environments sampled for the paper (8/12), it also leads to dramatically worse performance in some environments. Thus, the novelty and effectiveness of the approach is limited.
> >
>
> In the new version of the paper, we updated the scores for the locomotion tasks, now the proposed method outperforms other latent-policy approaches on the whole set. The issue was the use of three layers for the policy network as opposed to typically utilized four layers in other algorithms.
>
> > This paper criticizes prior work for having to manually bound the RL agent's action space with the VAE approach, but also takes the approach of manually bounding the agent's actions to be within the (-1,+1) interval of support of the uniform distribution. So it would be better to tone down or rephrase those claims. For example, the related work talks about how CNF "makes agents conservative by design, rather than squeezing policy outputs". But in effect you are squeezing policy outputs also.
> >
>
> We agree that the highlighted criticism might have sound a little bit unfair, so we removed it. Overall, we wanted to emphasize the issue of *manually* choosing the clipping constant (e.g., for PLAS, the squeezing interval is actually picked different for each environment). So that the aim of the study was to demonstrate that the proposed architecture allows for better modelling of the latent space (e.g., as demonstrated by better scores) and avoids the need for manual clipping, which we, hopefully, further bolstered by more experiments in the updated version of the paper demonstrating that CNF outperforms NF-Normal with different clipping values.
>
> > It appears the paper tests on a subset of the D4RL environments. Why were these chosen vs. others? This should be justified better.
> >
>
> We chose this subset of environments as this set of datasets seems to be universally used for testing new offline RL algorithms (some papers, indeed, use more, but with the limited computational budget we wanted to put resources on the ablations).
>
> > However, the top paragraph of p. 8 states that the ablation using a Normal latent space was carried out with the best hyperparameters found for the Uniform latent space. So how do we know that the difference in performance can't be attributed to doing better hyperparameter tuning for the proposed approach?
> >
>
> We added more discussion on this choice in the appendix. Overall, one can see that the training and validation loss curves do coincide, which suggests that they posses similar encoding and reconstruction capacity.

---

### Decision · Program_Chairs · 2023-01-20

**Decision:**

Reject

**Justification For Why Not Higher Score:**

See meta-review.

**Justification For Why Not Lower Score:**

n/a

**Metareview: Summary, Strengths And Weaknesses:**

The paper presents using normalizing flows rather than VAEs to constrain RL exploration.  The idea is simple and seems to work, but is very incremental.  The improvement over existing literature is minimal cq. inconsistent.

Unfortunately the authors took only minimal use of the rebuttal phase.

**Summary Of Ac-Reviewer Meeting:**

See above.